# Contrast-Enhanced Intraoperative Ultrasound Shows Excellent Performance in Improving Intraoperative Decision-Making

**DOI:** 10.3390/life14091199

**Published:** 2024-09-22

**Authors:** Laura S. Kupke, Ivor Dropco, Markus Götz, Paul Kupke, Friedrich Jung, Christian Stroszczynski, Ernst-Michael Jung

**Affiliations:** 1Department of Radiology, University Hospital Regensburg, 93053 Regensburg, Germany; christian.stroszczynski@ukr.de (C.S.); ernst-michael.jung@ukr.de (E.-M.J.); 2Department of Surgery, University Hospital Regensburg, 93053 Regensburg, Germany; ivor.dropco@ukr.de (I.D.); markus.goetz@ukr.de (M.G.); paul.kupke@ukr.de (P.K.); 3Institute of Biotechnology, Brandenburg University of Technology Cottbus-Senftenberg, 01968 Senftenberg, Germany; friedrich.jung@b-tu.de

**Keywords:** ultrasound, contrast-enhanced ultrasound, CEUS, micro-vascularization, intraoperative ultrasound, IOUS, intraoperative decision-making

## Abstract

Background: The aim of this study was to evaluate the performance and the impact of contrast-enhanced intraoperative ultrasound (CE-IOUS) on intraoperative decision-making, as there is still no standardized protocol for its use. Therefore, we retrospectively analyzed multiple CE-IOUS performed in hepato-pancreatic-biliary surgery with respect to pre- and postoperative imaging and histopathological findings. Methods: Data of 50 patients who underwent hepato-pancreatic-biliary surgery between 03/2022 and 03/2024 were retrospectively collected. CE-IOUS was performed with a linear 6–9 MHz multifrequency probe connected to a high-resolution device. The ultrasound contrast agent used was a stabilized aqueous suspension of sulphur hexafluoride microbubbles. Results: In total, all 50 lesions indicated for surgery were correctly identified. In 30 cases, CE-IOUS was used to localize the primary lesion and to define the resection margins. In the remaining 20 cases, CE-IOUS identified an additional lesion. Fifteen of these findings were identified as malignant. In eight of these cases, the additional malignant lesion was subsequently resected. In the remaining seven cases, CE-IOUS again revealed an inoperable situation. In summary, CE-IOUS diagnostics resulted in a high correct classification rate of 95.7%, with positive and negative predictive values of 95.2% and 100.0%, respectively. Conclusions: CE-IOUS shows excellent performance in describing intraoperative findings in hepato-pancreatic-biliary surgery, leading to a substantial impact on intraoperative decision-making.

## 1. Introduction

Due to its noninvasiveness and its broad availability, as well as its easy application, ultrasound usually is the first approach in imaging of abdominal tumors, especially liver tumors [1,2,3]. Additionally, the increasing use of AI in medicine in general and also in sonography shows that ultrasound will play a significant role in the future through improved evaluation options [4,5]. In the subsequent diagnostic course, usually computed tomography (CT) or magnet resonance imaging (MRI) scans are performed to complement imaging [6]. Before the decision for surgery is made, all existing examinations are generally considered in an interdisciplinary tumor board. When surgery is conducted, the localization of the tumor can be estimated according to preoperative imaging. Extended tumors or subcapsular liver lesions can regularly be palpated by the surgeon, whereas small or deep lesions are difficult to detect without tools [7]. For the exact assessment of the tumor size and its interaction with the surrounding tissue and vessels, intraoperative ultrasound (IOUS) is indispensable, as preoperative ultrasound shows limitations due to percutaneous and angled application [8]. IOUS, in contrary, is conducted directly on the organ surface. Hence, special linear multifrequency probes are used [9]. As even high resolution B-mode techniques often fail to detect the lesions in their whole expansion, contrast-enhanced IOUS (CE-IOUS) is used [8]. Therefore, an intravenous ultrasound contrast agent (UCA) is applied to help differentiate malignant and benign lesions by visualizing the microcirculation. The UCA used is a suspension containing gas microbubbles which are exhaled by the patient a few minutes after injection. It therefore leads to less side effects for the patient compared to contrast agents used for CT or MRI [10]. The UCA has the ability to remain intravascular, thus enabling dynamic imaging of a lesion with tumor vessels and its surrounding microvascularization [11,12]. In general, knowledge of tumor microvascularization is necessary for effective and successful treatment, especially to guarantee adequate surgical margins [13,14]. Typical criteria for malignancy in CE-IOUS are surrounding neovascularization, irregular wall side hypervascularization, and central washout or hypoperfusion. Benign lesions show rather organized hypervascularization from the wall towards the center [15]. With this knowledge, suspect lesions can be defined for surgical resection, whereas visually benign lesions are left in situ. Overall, CE-IOUS provides information that helps to assess surgical resectability and therefore has direct impact on patient outcomes regarding complications and long-term survival. Several studies have demonstrated the beneficial implementation of CE-IOUS but there is no standardized protocol for its use [16,17,18,19,20].

In this study, we retrospectively analyzed several CE-IOUSs performed in hepato-pancreatic-biliary surgery to evaluate the performance and the impact on intraoperative decision-making, as it is still not yet part of standardized patient care.

## 2. Materials and Methods

### 2.1. Data Collection

The aim of this retrospective study was to evaluate the performance and the impact of CE-IOUS on intraoperative decision-making. Therefore, data were collected on 50 patients who underwent hepato-pancreatic-biliary surgery between 03/2022 and 03/2024. Inclusion criteria were pre- and postoperative imaging with either CT or MRI, suspected lesion in the hepato-pancreatic-biliary tract, and potential resectability. Pre- and postoperative CT or MRI scans, surgical reports, CE-IOUS findings, and histopathological reports were available for each case. Patients’ treatment was managed according to local guidelines.

The acquired data was analyzed using GraphPad Prism v10 (Boston, MA, USA) and statistics were presented as indicated in the respective tables and figures. 

The study was approved by the local ethics committee (approval number 18-1137-104).

### 2.2. Conduction of CE-IOUS

The ultrasound device used for all intraoperative examinations was a high-resolution device (LOGIQ E9, GE Healthcare, Chicago, IL, USA) equipped with a linear multifrequency probe (6–9 MHz). CE-IOUS was indicated interdisciplinarily by the surgeons and the radiologists. In the operation room, the ultrasound probe was then wrapped sterilely and the patient was registered in the system. During all intraoperative examinations, the surgeon guided the ultrasound probe on the patient and the radiologist performed optimizations on the device and evaluated the imaging. First, an examination in fundamental B-mode and vascularization supplemented by color-coded duplex sonography was completed. In this study, all CE-IOUSs were performed by the same experienced attending radiologist with a DEGUM III certificate. The UCA used for CE-IOUS was a stabilized aqueous suspension of microbubbles composed of Sulphur hexafluoride (SonoVue^®^, Bracco, Italy) resulting in an increased backscatter of ultrasound. Due to the small size of the microbubbles (<10 µm) and the ability to remain intravascular, the dynamic micro-vascularization of lesions can be visualized. A total of 2.4–4.8 mL of UCA were administered intravenously to perform CE-IOUS. Firstly, the primary lesion was localized and characterized. Afterwards, a scan of the whole organ and surrounding tissue was performed to detect potential additional lesions and define the resection margin. Surrounding neovascularization, irregular wall side hypervascularization, and central washout or hypoperfusion were considered as criteria for malignancy. Regarding the contrast phases after UCA application, the early arterial phase after 10 to 15 s can be distinguished from the portal venous phase beginning after 60 s.

### 2.3. Analysis of CE-IOUS Accuracy and Impact

The CE-IOUS results were compared to pre- and postoperative imaging and intra- and postoperative histopathological findings to assess the accuracy of CE-IOUS. To analyze the impact of CE-IOUS on intraoperative decision-making, the surgical reports and the CE-IOUS findings were evaluated. Firstly, the cases were divided according to whether CE-IOUS was only used to define the resection margins of the tumor or if CE-IOUS also identified additional lesions. This subgroup was split further whether the finding was categorized as benign or malignant. The lesions categorized as malignant were additionally analyzed in case they resulted in operative resection or stop of surgery. According to this division, we quantified the impact on surgery procedure with two subgroups: moderate and fundamental. Moderate impact was achieved when CE-IOUS was used to define the resection margins, whereas fundamental impact was associated with the identification of additional lesions.

## 3. Results

### 3.1. Study Cohort

The patient cohort consisted of 28 males and 22 females with a mean age of 60.66 years (95% CI 56.17–65.15). The mean lesion diameter was 3.27 cm (95% CI 2.53–4.02). The mean time span between the latest preoperative imaging and surgery was 23.62 days (95% CI 15.15–31.25). The postoperative histopathology results showed 18 liver metastases (36%), 9 hepatocellular carcinomas (18%), 9 cholangiocellular carcinomas (18%), 8 pancreatic carcinomas (16%), 3 benign diseases (6%), 2 hepatoblastomas (4%) and 1 duodenal carcinoma (2%) (Table 1).

### 3.2. Impact on Intraoperative Decision-Making

To evaluate the impact on intraoperative decision-making, the cases included in this study were categorized into subgroups shown in Figure 1. In 30 cases, CE-IOUS was used to localize the primary lesion and to define the resection margins (Figure 2 and Figure 3). In the remaining 20 cases, in addition to defining the resection margins, an additional lesion was identified by CE-IOUS (Figure 4). In total, 15 of these findings were marked as malignant. In eight of these cases, the additional malignant finding was resected consecutively. In the remaining seven cases, CE-IOUS revealed an inoperable situation. The background to these seven cases is shown in Table 2.

Applying the categorization we introduced, moderate impact on the procedure of surgery was achieved in 60.0% of cases with sole definition of resection margins. In the remaining 40.0%, CE-IOUS identified additional lesions and therefore led to fundamental impact on intraoperative decision-making; in 16.0% of cases, additional resection was conducted and in 14.0%, surgery had to be stopped without any resection. In the remaining 10.0%, lesions were graded as benign and therefore left in situ (Figure 5).

### 3.3. Accuracy of CE-IOUS

In this study, all 50 surgery-indicating lesions were correctly identified. Three of the additional identified lesions were falsely marked as malignant. Histopathological analysis described one finding as a non-tumor-infiltrated lymph node. In the two other cases, the lesions were graded as benign. In summary, diagnostics by CE-IOUS led to a high correct classification rate (CCR) of 95.7% with a positive and negative predictive value of 95.2% and 100.0% in this study, respectively (Figure 6a). Analyzing the additional lesions separately, a CCR of 85.0% with a positive and negative predictive value of 80.0% and 100.0% was achieved in this study (Figure 6b).

## 4. Discussion

CE-IOUS is a dynamic technique with high resolution down to the capillary level. Main areas of application include the precise classification of already known lesions and the detection of lesions that are difficult to recognize in other modalities [17]. It helps to detect lesions and their microvasculature from the early arterial phase to a late phase up to 5 min [21]. Often, the lengthwise tumor expansion and depth, as well as the tumor margins, are complex to verify in B-mode ultrasound. Thus, complete resection with tumor free margins (R0) can be difficult to achieve. Using CE-IOUS, margins can be detected more reliably due to typical wash-out of malignant lesions and peritumoral neovascularization [16,17,22,23]. Westwood et al. showed in a large analysis that CEUS provided equal diagnostic power in characterizing liver lesions compared to CT or MRI [24]. Seitz et al. demonstrated similar results when comparing CEUS with MRI [25]. However, the intraoperative use of ultrasound should not replace CT and MRI, but rather be used to gain additional information and to identify further lesions.

In this study, we analyzed the accuracy of CE-IOUS and its impact on intraoperative decision-making. Our study has limitations due to its retrospective design as we had to rely on the completeness and correctness of the available results.

It must be emphasized that in this study we analyzed a heterogeneous patient population with mainly primary (20 cases) and secondary (18 cases) liver tumors, but also multiple pancreatic adenocarcinomas (8 cases) in order to get an overall impression of the CE-IOUS performance. Although studies have shown similar performance rates of CE-IOUS for liver and pancreatic lesions, contrast behavior and surgery techniques differ between those two groups [19,26,27,28].

CE-IOUS resulted in a high CCR of 95.7%, with positive and negative predictive values of 95.2% and 100.0%, respectively. Similar rates have been reported in other studies [20,29,30]. In addition to the correct localization of the primary lesion in all cases included, our study showed a high rate of additional lesions, identified in 20 of 50 cases. This is in line with other studies reporting high frequencies of additional lesions during IOUS of up to 50% [18,26]. Overall, the high CCR, together with a high amount of additional lesions confirms the hypothesis that CE-IOUS is crucial in providing additional information in comparison to pre-surgery CT or MRI [31,32]. The additional findings led to a change in the planned procedure during surgery in most cases, which had a direct impact on patient outcomes in terms of recurrence rate, morbidity and mortality.

However, as ultrasound is a dynamic procedure, it is necessary that it is performed by experienced sonographers in addition to using high-resolution equipment to maintain high standards [8]. In addition, the comparability between preoperative ultrasound and IOUS is limited due to varying examiners, but also due to different execution—percutaneous and angled versus directly on the organ surface. Prospective studies should investigate the standardization of this process.

The contrast agent used in this study was SonoVue^®^, a stabilized aqueous suspension of sulphur hexafluoride microbubbles. It is the most commonly used UCA worldwide and is associated with high efficacy and safety [10]. Sonazoid is another UCA with similar safety, consisting of perfluorobutane microspheres stabilized by a membrane of hydrogenated egg phosphatidylserine sodium. Currently, Sonazoid has approval only in a few countries worldwide, though it has the advantage of an additional contrast phase known as the Kupffer phase as it can be phagocytosed by Kupffer cells in the liver [33]. Several studies have shown the benefit of Sonazoid in detecting small liver lesions and HCC resulting in higher accuracy and sensitivity [34,35]. Other studies, however, could not confirm these results and showed comparable results between the two UCAs or even higher sensitivity and specificity for SonoVue^®^ [36,37,38].

## 5. Conclusions

CE-IOUS, as an important interdisciplinary tool, shows high accuracy in describing and grading intraoperative findings in hepato-pancreatic-biliary surgery, which has a crucial impact in intraoperative decision making and should therefore be implemented in the standardized surgical procedure.

## Figures and Tables

**Figure 1 life-14-01199-f001:**
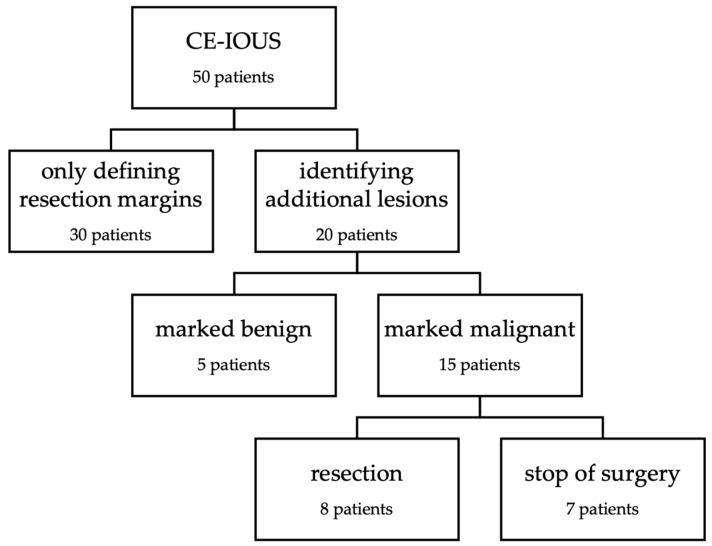
Flowchart demonstrating the categorization into subgroups after performance of contrast-enhanced intraoperative ultrasound (CE-IOUS).

**Figure 2 life-14-01199-f002:**
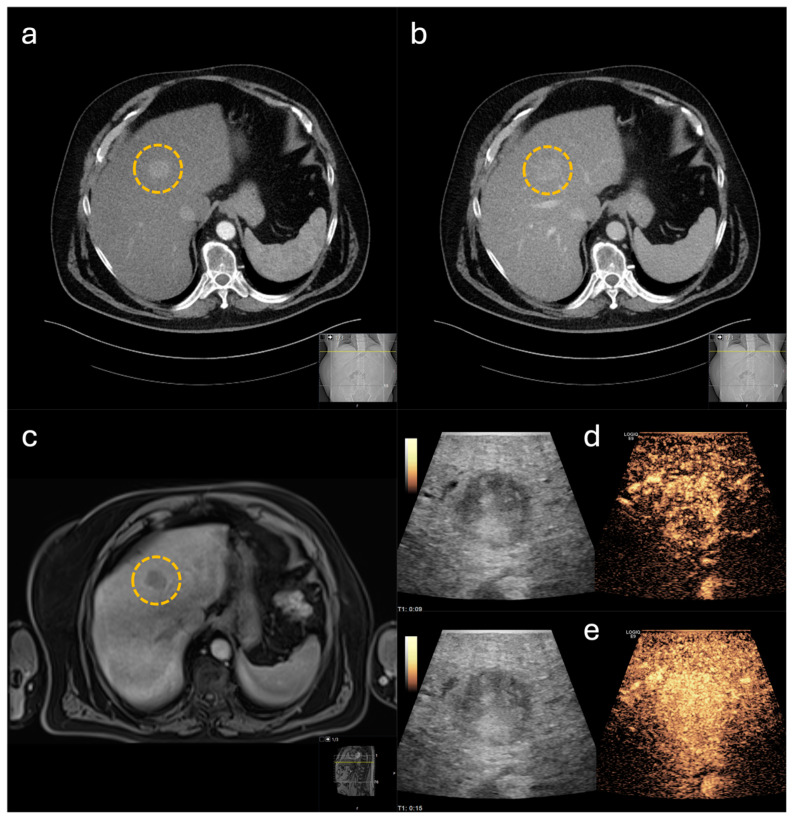
Hepatocellular carcinoma, marked by yellow circle, with typical contrast behavior in imaging: (**a**) computed tomography (CT) with hypervascularization in the arterial phase, (**b**) CT with washout in the portal-venous phase, (**c**) magnet resonance imaging with washout after contrast agent application, (**d**) contrast-enhanced intraoperative ultrasound (CE-IOUS) with surrounding neovascularization in early arterial contrast phase, (**e**) CE-IOUS with hypervascularization in the further course.

**Figure 3 life-14-01199-f003:**
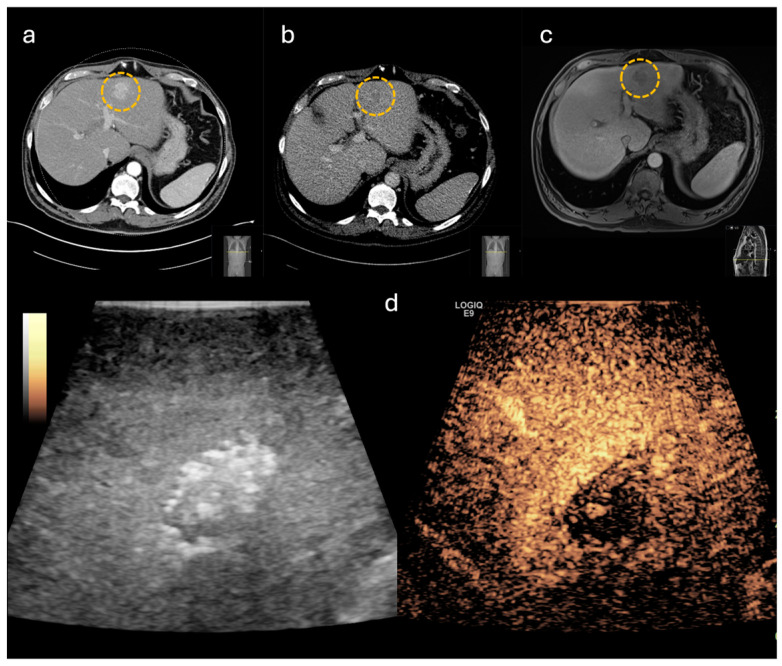
Hepatocellular carcinoma, marked by yellow circle, with typical contrast behavior in imaging. (**a**) computed tomography (CT) with hypervascularization in the arterial phase, (**b**) CT with washout in the portal venous phase, (**c**) magnet resonance imaging with washout after contrast agent application, (**d**) contrast-enhanced intraoperative ultrasound with central washout in the portal venous phase.

**Figure 4 life-14-01199-f004:**
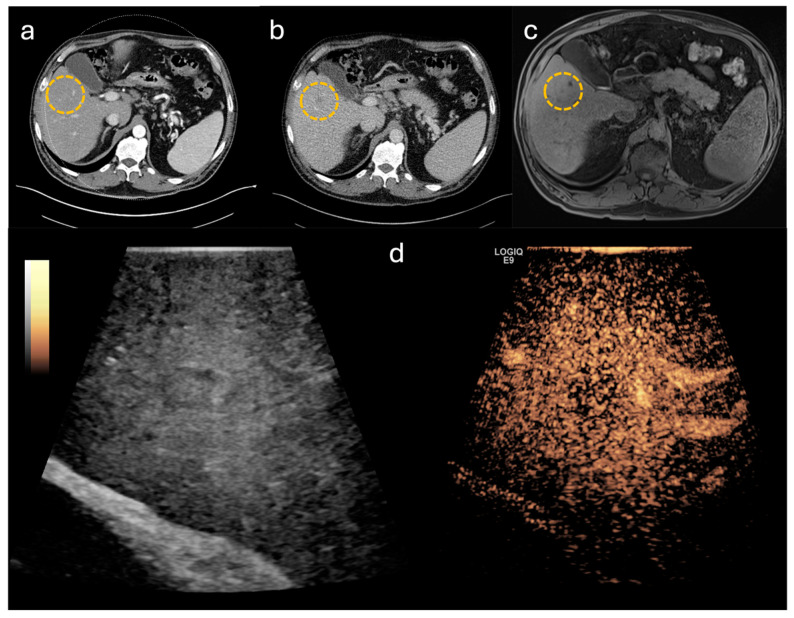
Unclear lesion, marked by yellow circle, in preoperative imaging: (**a**) computed tomography (CT) with homogenous liver parenchyma in arterial phase, (**b**) CT with hypodense lesion in portal-venous phase, (**c**) magnet resonance imaging with dull washout after contrast agent application, (**d**) contrast-enhanced intraoperative ultrasound with central washout in portal venous phase.

**Figure 5 life-14-01199-f005:**
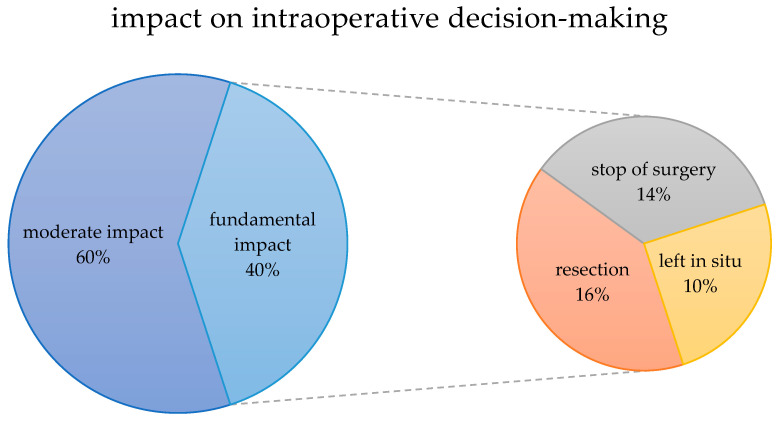
Categorized visualization of the impact of contrast-enhanced intraoperative ultrasound on intraoperative decision-making.

**Figure 6 life-14-01199-f006:**
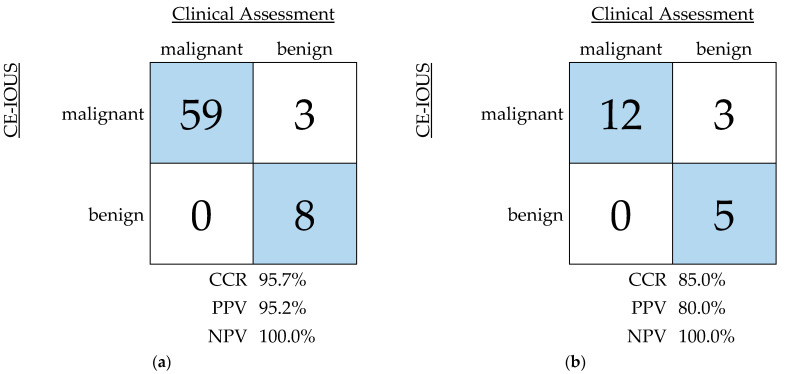
Predictive values of contrast-enhanced intraoperative ultrasound (CE-IOUS) in this study: (**a**) all lesions described by CE-IOUS; (**b**) additional lesions analyzed separately.

**Table 1 life-14-01199-t001:** General characteristics of the study cohort.

Patient Demographics	
Mean Age (years) [95% CI]	60.66 [56.17–65.15]
Sex Distribution ♂/♀	28/22
Mean Lesion Diameter (cm) [95% CI]	3.27 [2.53–4.02]
Mean Time Span from Imaging to Surgery (days) [95% CI]	23.20 [15.15–31.25]
Surgery-Indicating Diagnosis	
Liver Metastases, *n* (%)	18 (36.0)
Hepatocellular Carcinoma, *n* (%)	9 (18.0)
Cholangiocellular Carcinoma, *n* (%)	9 (18.0)
Pancreatic Carcinoma, *n* (%)	8 (16.0)
Benign Disease, *n* (%)	3 (6.0)
Hepatoblastoma, *n* (%)	2 (4.0)
Duodenal Carcinoma, *n* (%)	1 (2.0)

**Table 2 life-14-01199-t002:** Background of inoperable situation detected by contrast-enhanced intraoperative ultrasound (CE-IOUS).

Patient	Surgery-Indicating Diagnosis	Preoperative Imaging	Latest Imaging to Surgery (Days)	Cause of Surgery Stop Detected by CE-IOUS
1	Hepatocellular Carcinoma	CT and MRI *	74	Vessel infiltration
2	Hepatocellular Carcinoma	CT and MRI *	27	Prior unknown liver metastases
3	Cholangiocellular Carcinoma	CT and MRI *	26	Vessel infiltration
4	Cholangiocellular Carcinoma	CT and MRI	48	Vessel infiltration
5	Pancreatic Carcinoma	CT and MRI *	7	Prior unknown liver metastases
6	Pancreatic Carcinoma	CT and MRI	22	Vessel infiltration
7	Liver Metastases (Colorectal Carcinoma)	CT and MRI	22	Vessel infiltration

Note: MRI * indicates hepatobiliary contrast agent.

## Data Availability

All data presented within this manuscript are available upon request from the corresponding author.

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
