# Peer review of "Contrast-Enhanced Intraoperative Ultrasound Shows Excellent Performance in Improving Intraoperative Decision-Making"

_life, 2024, doi:10.3390/life14091199_

Round 1
Reviewer 1 Report
Comments and Suggestions for Authors
I have some questions for the authors, namely:
Abstract
1. Context. What is known in the literature about this subject? The purpose of the study should be included in the sub-chapter Methods!
2. To be clarified in comparison with what is this examination more efficient? In comparison with preoperative contrast ultrasound?
3. If the ultrasonography was performed preoperatively and intraoperatively by only one doctor or more for all 50 patients?, considering that the interpretation and visualization of the lesions depends a lot on the expertise of the examiner!
4. What does this study add compared to other studies in the literature?
5. Methods - Row 81-83 Was the ultrasound performed by the same examiner?? And if the examinations were with preoperative contrast, considering the role of contrast agents and the examiner in highlighting the lesions?
To be improved in Discussions and Conclusions
It should be emphasized that CE-IOUS has an important role in 2 main situations: the differentiation of unidentified lesions detected in cirrhotic liver by conventional IOUS and in the detection of colorectal liver metastases that can be overlooked by other imaging modalities.
Intraoperative performance of CE-IOUS directly affects surgical decision-making, which can significantly affect patient outcomes, see cited art (Chang GY, Fetzer DT, Porembka MR. Contrast-Enhanced Intraoperative Ultrasound of the Liver. Surg Oncol Clin N Am. 2022 Oct;31(4):707-719. doi: 10.1016/j.soc.2022.06.007. Epub 2022 Sep 27. PMID: 36243503.
It does not replace other imaging investigations, such as CT or MRI, but intraoperative use to identify liver and pancreatic lesions should be encouraged.
Intraoperative CE-IOUS limits. Of course, if the ultrasonography with contrast was performed preoperatively by the same examiner, the results might have been different. But prospective studies in several centers could clarify if a standardization of this procedure is necessary.
The bibliography is appropriate, I do not suggest other bibliographic sources.
Reviewer 2 Report
Comments and Suggestions for Authors
This original article by Kupke at al. aims to evaluate the performance and impact of CE-IOUS on decision-making during hepato-pancreatic-biliary surgeries. It highlights the significant contribution of CE-IOUS in accurately identifying lesions and influencing surgical outcomes, with a correct classification rate of 95.7%. The study’s strengths include its robust data set and clear demonstration of CE-IOUS’s potential to improve intraoperative precision. However, its retrospective design and reliance on the completeness of available data may limit the generalizability of the findings.
Overall, the paper is well written and organized. I have three (minor) comments:
1. The introduction is well-structured, but it could be further strengthened by including references to other similar studies that have evaluated the role of CE-IOUS in intraoperative decision-making.
2. While the paper presents detailed findings, it could benefit from providing more explicit criteria for how lesions were deemed malignant or benign. This might help other researchers replicate the study. Additionally, including more detailed statistical methods in this section would strengthen the rigor of the study.
3. The retrospective nature of the study is mentioned, but it would be valuable to expand the limitations section. For example, the authors could discuss how potential biases in data selection or the variability in surgeon or radiologist experience might have impacted results.
Comments on the Quality of English Language
Minor editing required, few grammatical errors.
Round 2
Reviewer 1 Report
Comments and Suggestions for Authors
I believe that the article can be published, the authors have increased the quality and clarity of this manuscript. The authors have clarified and provided a punctual answer to each requirement.